# OCTAVE GRAPH CONVOLUTIONAL NETWORK

## ABSTRACT

Many variants of Graph Convolutional Networks (GCNs) for representation learning have been proposed recently and have achieved fruitful results in various domains. Among them, spectral-based GCNs are constructed via convolution theorem upon a theoretical foundation from the perspective of Graph Signal Processing (GSP). However, despite most of them implicitly act as low-pass filters that generate smooth representations for each node, there is limited development on the full usage of underlying information from low-frequency components. Here, we first introduce the octave convolution on graphs in spectral domain. Accordingly, we present Octave Graph Convolutional Network (*OctGCN*), a novel architecture that learns representations for different frequency components regarding weighted filters and graph wavelets bases. We empirically validate the importance of low-frequency components in graph signals on semi-supervised node classification and demonstrate that our model achieves state-of-the-art performance in comparison with both spectral-based and spatial-based baselines.

## 1 INTRODUCTION

The family of Graph Convolutional Networks (GCNs) (Zhang et al., 2018), which generalizes the traditional Convolutional Neural Networks (CNNs) from Euclidean structure data to graphs, has achieved a remarkable success in various application domains, including but not limited to social networks (Chen et al., 2018), computer vision (Kampffmeyer et al., 2018), text classification (Yao et al., 2019) and applied chemistry (Liao et al., 2019).

Existing methods of GCNs design falls into two categories: spatial-based methods and spectral-based methods (Wu et al., 2019). On the surface, the spatial-based models directly perform information aggregation through graph topology. However, this aggregation can be viewed as a simplified convolution operation on spectral domain with the theoretical foundation in Graph Signal Processing (GSP). GSP extends the concepts in Discrete Signal Processing (DSP) and focuses on analyzing and processing data points whose relations are modeled as graph (Shuman et al., 2013; Ortega et al., 2018). In standard signal processing problems, the underlying "real signal" is usually assumed to have low frequencies (Rabiner & Gold, 1975). Recent works (Wu et al., 2019; Maehara, 2019) reveal that the spectral-based GCNs can be viewed as an implicit low-pass-type filter based denoising mechanism on the spectral domain. However, there is still a lack of the explicit learning architecture of GCNs to extract the beneficial information from low-frequency while making full use of the high-frequency under certain scenarios.

Considering the signal processing problem in computer vision, a natural image can be decomposed into a low spatial frequency component containing the smoothly changing structure, *e.g.*, background, and a high spatial frequency component describing the rapidly changing fine details, *e.g.*, outlines. To accommodate with this phenomenon, (Chen et al., 2019) proposed Octave Convolution (OctConv) to learn the octave feature representations, which factorizes convolutional feature maps into two groups at different spatial frequencies and process them with different convolutions at their corresponding frequency. Similarly, the octave mechanism is observed in graph representational learning more naturally. The eigenvectors associated with small eigenvalues carry smoothly varying signal, encouraging nodes that are neighbors to share similar values. In contrast, the eigenvectors associated with large eigenvalues carry sharply varying signal across edges (Donnat et al., 2018). Accordingly, extending octave convolution from images to graphs sheds light on the explicit learning of GCNs regarding the representation of different frequencies.

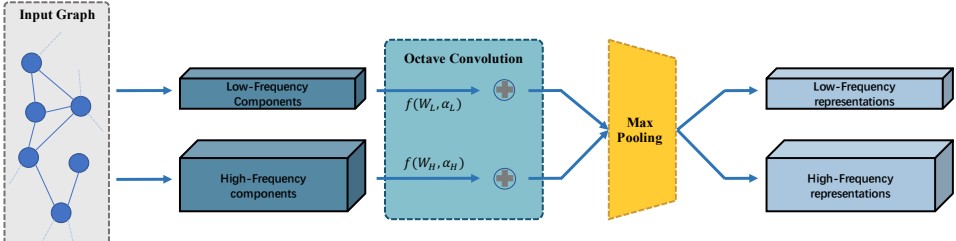

Figure 1: The overview of octave convolutional learning on graphs in spectral domain.

Different from the scale-space theory (Lindeberg, 2013) utilized in OctConv (Chen et al., 2019) to define the low- and high-frequency spaces, graph signal processing (GSP) provides us a way to directly divide the low- and high-frequency components based on the ascending ordered eigenvalues of Laplacian. Inspired from this, we propose to consider the octave feature in the spectral domain to construct a new graph convolutional model: Octave Graph Convolutional Network *OctGCN*. In *OctGCN*, with a particular design of filters for different spectrum, we allocate different weights on low- and high-frequency. Spectral graph wavelets are chosen as feature transformation bases due to their local and sparse property. Two parameters are further introduced to construct the filters for reducing the parameter complexity to the same as (Kipf & Welling, 2017), which is critical when labels of training data are limited. Meanwhile, we employ the *attention* mechanism to learn the importance of low and high pass filters. Figure 1 provides the overview of the design of *OctGCN* in spectral domain. We validate the effectiveness of our model via experiments on semi-supervised node classification tasks, where the expressive power of GCNs is crucial to capture the underlying beneficial information in graph signals. Our results confirm the importance of low-frequency in graphs and bring interpretability to the innate character of GCNs. In addition, empirical results show that our proposed method consistently rivals the state-of-art methods from both spectral-based and spatial-based baselines on real-world datasets.

## 2 RELATED WORK

**Spectral convolutional networks on graphs.** Existing methods of defining a convolutional operation on graphs can be broadly divided into two categories: spectral based and spatial based methods (Zhang et al., 2018). We focus on the spectral graph convolutions in this paper. Spectral CNN (Bruna et al., 2014) first attempts to generalize CNNs to graphs based on the spectrum of the graph Laplacian and defines the convolutional kernel in the spectral domain. (Boscaini et al., 2015) further employs windowed Fourier transformation to define a local spectral CNN approach. ChebyNet (Defferrard et al., 2016) introduces a fast localized convolutional filter on graphs via Chebyshev polynomial approximation. Vanilla GCN (Kipf & Welling, 2017) further extends the spectral graph convolutions considering networks of significantly larger scale by several simplifications. (Khasanova & Frossard, 2017) learns graph-based features on images that are inherently invariant to isometric transformations. Cayleynets (Levie et al., 2018) alternatively introduce Cayley polynomials allowing to efficiently compute spectral filters on graphs. Lanczos algorithm is utilized in LanczosNet (Liao et al., 2019) to construct low-rank approximations of the graph Laplacian for convolution. SGC (Wu et al., 2019) further reduces the complexity of Vanilla GCN by successively removing the non-linearities and collapsing weights between consecutive layers. Despite their effective performance, all these convolution theorem based methods lack the strategy to explicitly treat low- and high-frequency components with different importance.

**Spectral graph wavelets.** Theoretically, the lifting scheme is proposed for the construction of wavelets that can be adapted to irregular graphs in (Sweldens, 1998). (Hammond et al., 2011) defines wavelet transforms appropriate for graphs and describes a fast algorithm for computation via fast Chebyshev polynomial approximation. For applications, (Tremblay & Borgnat, 2014) utilizes graph wavelets for multi-scale community mining and obtains a local view of the graph from each node. (Donnat et al., 2018) introduces the property of graph wavelets that describes information diffusion and learns structural node embeddings accordingly. GWNN (Xu et al., 2019a) first attempts to construct graph neural networks with graph wavelets. These works emphasize the local and sparse property of graph wavelets for graph signal processing both theoretically and practically.

**Octave feature representation.** In computer vision, (Chen et al., 2019) first defines octave feature representations based on scale-space theory and reduces spatial redundancy of vanilla CNN models. (Durall et al., 2019) further leverages octave convolutions for designing stabilizing GANs. To our knowledge, this is the first time that octave feature representations are considered in irregular graph domain and established with graph convolutional neural networks.

# 3 PROPOSED APPROACH

## 3.1 PRELIMINARY

We denote $\mathcal{G} = \{V, E\}$ as an undirected graph, where $|V| = n$ is the set of $n$ nodes, and $E$ is the set of edges. The adjacency matrix is defined as $A$ with $A_{i,j} = A_{j,i}$ describing the edge connecting node $i$ and node $j$. The graph Laplacian matrix $\mathcal{L}$ is defined as the difference $\mathcal{L} = D - A$, where $D_{i,i} = \sum_j A_{i,j}$ is a diagonal degree matrix. The normalized graph Laplacian matrix is referred as $L = I_n - D^{-1/2} A D^{-1/2}$, where $I_n$ is the identity matrix. The graph Laplacian $L$ can be decomposed into its eigenvalue components, $L = U \Lambda U^\top$, such that for the set of eigenvalues in ascending order $\{\lambda_i\}_{i=0}^{n-1} = \lambda_0 \leq \lambda_1 \leq \cdots \leq \lambda_{n-1}$, the diagonal eigenvalue matrix is defined as $\Lambda = \text{diag}(\lambda_0, \ldots, \lambda_{n-1})$ and $U = (\boldsymbol{u}_1, \boldsymbol{u}_2, ..., \boldsymbol{u}_n)$ is the eigenvector matrix.

Since $L$ is a real symmetric matrix, it has real, non-negative eigenvalues $\{\lambda_i\}_{i=0}^{n-1} = \lambda_0 \leq \lambda_1 \leq \cdots \leq \lambda_{n-1}$, known as the **frequencies** of graph. These eigenvalues have associated a complete set of orthonormal eigenvectors in $U$, identified as Laplacian eigenvectors. In Graph Signal Processing (GSP), we denote frequency components with small/large eigenvalues of Laplacian as **low/high** frequencies. Given a signal $\boldsymbol{x}$ and a graph Laplacian $L$, the Graph Fourier Transform (GFT) of $\boldsymbol{x}$ with respect to $L$ is defined as the signal $\tilde{\boldsymbol{x}} = U^\top \boldsymbol{x}$, and the inverse (i)GFT of $\boldsymbol{x}$ with respect to $L$ is $\boldsymbol{x} = U \tilde{\boldsymbol{x}}$ (Shuman et al., 2013).

## 3.2 SPECTRAL GRAPH CONVOLUTION

The spectral convolution on graphs is normally defined as the multiplication of a signal on every node $\boldsymbol{x}$ with a diagonal filter $g_\theta = \text{diag}(\theta)$ parameterized by $\theta$ in the Fourier domain:

$$g_\theta \star \boldsymbol{x} = U g_\theta U^\top \boldsymbol{x}, \tag{1}$$

The filters are usually understood as a function of the eigenvalues. Inspired by (Maehara, 2019), we can decompose the spectral convolution process on graphs from the perspective of GSP as four steps: **1.** Compute the graph bases $U$; **2.** Graph spectral transform on signal $\boldsymbol{x}$ with $U^\top$; **3.** Filtering with $g_\theta$; **4.** Reconstruct the signal features in the spatial domain with $U$. In this sense, the design of filter $g_\theta$ is essential to the performance of spectral convolution. Broadly, the filter design can be divided into two categories: it is either learned by the neural network (Bruna et al., 2014; Xu et al., 2019a) or directly fixed as the eigenvalues via approximation (Kipf & Welling, 2017; Wu et al., 2019). In this paper, we will focus on the first kind.

Spectral CNN (Bruna et al., 2014) generalizes the convolutional net by operating on the spectrum of weights, given by the ordered eigenvectors of its graph Laplacian. The structure of $k$-th layer is constructed as:

$$X_{[:,j]}^{k+1} = h\left(U \sum_{i=1}^{p} F_{i,j}^k U^\top X_{[:,i]}^k\right) \qquad j = 1, \cdots, q, \tag{2}$$

where $X^k \in \mathbb{R}^{n \times p}$ is the signal with $p$ input channels and $X^{k+1} \in \mathbb{R}^{n \times q}$ is the convolved signal matrix. $X_{[:,i]}^k$ and $X_{[:,j]}^{k+1}$ are the $i$-th and $j$-th column of $X^k$ and $X^{k+1}$, respectively. $g_\theta$ of $k$-th layer is defined as $F_{i,j}^k$, which is a diagonal filter matrix to be learned for each input channel in spectral domain. $h$ is a real valued nonlinear activation function, *e.g.*, $\text{ReLU}(\cdot) = \max(0, 1)$. Thus, the parameter complexity of Spectral CNN is $\mathcal{O}(n \times p \times q)$, which generally demands a huge amount of training data for parameter learning.

## 3.3 WHY SPECTRAL GRAPH WAVELET?

Graph wavelet neural network (GWNN) (Xu et al., 2019a) expands the spectral convolution from Fourier transformation to wavelet transformation. Let $g_s(\lambda) = e^{-\lambda s}$ be a heat kernel filter with

scaling parameter $s$. In GSP (Hammond et al., 2011; Shuman et al., 2013), the spectral graph wavelet $\psi_{si}$ is defined as the signal resulting from the modulation in the spectral domain of a signal $\boldsymbol{x}$ centered around the associated node $i$. Then, the graph wavelet transform is conducted by employing a set of wavelets $\psi_s = (\psi_{s1}, \psi_{s2}, \ldots, \psi_{sn})$ as bases. Formally, the spectral graph wavelets are given as:

$$\psi_s = U g_s U^\top, \tag{3}$$

where $U$ is Laplacian eigenvectors of $L = D - A$ or normalized Laplacian $L = I_n - D^{-\frac{1}{2}} A D^{-\frac{1}{2}}$, $g_s = \mathrm{diag}\big(g_s(\lambda_1), g_s(\lambda_2), \ldots, g_s(\lambda_n)\big)$ is a scaling matrix with heat kernel. The inverse of graph wavelets $\psi_s^{-1}$ is obtained by simply replacing the $g_s(\lambda)$ in $\psi_s$ with $g_s(-\lambda)$ corresponding to the heat kernel (Donnat et al., 2018). Similarly, smaller indices in graph wavelets correspond to low-frequency components and vice versa.

Similar to GFT, after replacing the Fourier bases with spectral graph wavelets, the graph wavelet transformation of a signal x on graph is defined as $\hat{\mathbf{x}} = \psi_s^{-1} x$ and the inverse graph wavelet transform is $x = \psi_s \hat{\mathbf{x}}$. Replacing the graph Fourier transform in spectral convolution (Equation 1) with graph wavelet transform, the graph wavelet convolution can be obtained as:

$$g_\theta \star \boldsymbol{x} = \psi_s g_\theta \psi_s^\top \boldsymbol{x} \tag{4}$$

The benefits that spectral graph wavelet bases have over Fourier bases mainly fall into two aspects: **1.** Given the sparse real-world networks, the graph wavelet bases are usually much more sparse than Fourier bases, *e.g.*, the density of $\psi_s$ is $2.8\%$ comparing with $99.1\%$ of $U$ (Xu et al., 2019a). The sparseness of graph wavelets makes them more computationally efficient for use. **2.** In spectral graph wavelets, the signal $\psi_s$ resulting from heat kernel filter $g_s$ is typically localized on the graph and in the spectral domain (Shuman et al., 2013). By adjusting the scaling parameter $s$, one can easily constrain the range of localized neighborhood. Smaller values of $s$ generally associate with smaller neighborhoods.

Employing the same strategy as in Spectral CNN (Bruna et al., 2014), GWNN designs the same diagonal filter $F_{i,j}^k$ to be learned for each input channel. The structure of $k$-th layer of GWNN is:

$$X_{[:,j]}^{k+1} = h\Big(\psi_s \sum_{i=1}^{p} F_{i,j}^k \psi_s^{-1} X_{[:,i]}^k\Big) \qquad j = 1, \cdots, q, \tag{5}$$

Note that both Spectral CNN and GWNN employ the same filters for learning full frequency components. As mentioned before, the parameter complexity of Spectral CNN is large since each pair of input and output channel requires learning an individual diagonal filter matrix $F_{i,j}^k$. GWNN further reduces the parameter complexity by dividing each layer into two components: feature transformation and graph convolution:

$$\text{feature transformation}: \ X^{k'} = X^k W^k, \tag{6}$$

$$\text{graph convolution}: \ X^{k+1} = h\Big(\psi_s F^k \psi_s^{-1} X^{k'}\Big). \tag{7}$$

where $W^k \in \mathbb{R}^{p \times q}$ is the feature transformation parameter matrix similar to Vanilla GCN (Kipf & Welling, 2017). In such a way, the feature transformation operation is detached from graph convolution and the parameter complexity is decreased from from $\mathcal{O}(n \times p \times q)$ to $\mathcal{O}(n + p \times q)$.

### 3.4 Octave Convolutional Layer

In contrast to the scale-space theory-based octave feature representation utilized in computer vision (Lindeberg, 2013), Graph Signal Processing provides us a more principle way in the spectral domain. To better capture the different importance of low- and high- frequency components and combining the benefits of graph wavelets, we can naturally construct each layer in an octave convolution manner by learning two different filters as:

$$\text{feature transformation}: \ X_L^{k'} = X^k W_L^k, \ X_H^{k'} = X^k W_H^k \tag{8}$$

$$\text{graph convolution}: \ X_L^{k+1} = \psi_{sL} F_L^k \psi_{sL}^{-1} X_L^{k'}, \ X_H^{k+1} = \psi_{sH} F_H^k \psi_{sH}^{-1} X_H^{k'}, \tag{9}$$

where $F_L^k \in \mathbb{R}^{d \times d}$ and $F_H^k \in \mathbb{R}^{(n-d) \times (n-d)}$ are the diagonal filter matrix for graph convolution to be learned with different weights for low- and high- components, respectively. $d$ is the hyper-parameter

Table 1: The overview of dataset statistics.

| Dataset | Nodes | Edges | Classes | Features | Label rate |
|---------|-------|-------|---------|----------|------------|
| Citeseer | 3,327 | 4,732 | 6 | 3,703 | 0.036 |
| Cora | 2,708 | 5,429 | 7 | 1,433 | 0.052 |
| Pubmed | 19,717 | 44,338 | 3 | 500 | 0.003 |

to select the proportion $d/n$ of low frequency components. $\psi_{sL}$ and $\psi_{sL}$ are the corresponding low- and high- frequency graph wavelet bases. Further, with a pooling operation on the outputs and non-linear activation function, the structure of the $k$-th layer in our structure can be defined as

$$X^{k+1} = h\Big(\text{Pooling}(\psi_{sL}F_L^k\psi_{sL}^{-1}X^kW_L^k, \ \psi_{sH}F_H^k\psi_{sH}^{-1}X^kW_H^k)\Big) \qquad (10)$$

We refer this proposed architecture as Octave Graph Convolutional Network (*OctGCN*).

Since the parameters in the diagonal convolution filtering kernels could be huge especially for large graphs, the graph-based semi-supervised learning might prohibit the parameter learning due to the limited amount of training data. To mitigate this issue, we further reduce the parameter complexity by constructing the graph convolution kernel $F^k$ with two parameters $\alpha_L$ and $\alpha_H$, and keeping the same weight matrix $W$ shared between low- and high- frequency components as

$$X^{k+1} = h\Big(\text{Pooling}(\psi_{sL}\begin{bmatrix} \alpha_L & & \\ & \ddots & \\ & & \alpha_L \end{bmatrix}\psi_{sL}^{-1}X^kW^k, \ \psi_{sH}\begin{bmatrix} \alpha_H & & \\ & \ddots & \\ & & \alpha_H \end{bmatrix}\psi_{sH}^{-1}X^kW^k)\Big) \qquad (11)$$

For the learning of weights of low- and high-frequency components $\alpha_L$ and $\alpha_H$, we adopt the *attention* strategy to constraint them within the scale of $(0, 1)$:

$$\alpha_* = \text{softmax}(\alpha_*) = \frac{\exp(\alpha_*)}{\sum_* \exp(\alpha_*)}, \quad * = L, H$$

Hence, we introduce three more hyper-parameters to be tuned: $\alpha_L$ and $\alpha_H$ control the importance of low and high frequency components, and parameter $d$ specify the ratio of low frequencies we expect to represent the graph. In this way, we reduces the parameter complexity from $\mathcal{O}(n + p \times q)$ in GWNN (Xu et al., 2019a) to $\mathcal{O}(p \times q)$, which is the same as Vanilla GCN (Kipf & Welling, 2017).

### 3.5 Fast Spectral Graph Wavelet Approximation via Chebyshev Polynomials

Directly computing the transformation according to Equation 3 is intensive for large graphs, since diagonalizing Laplacian $L$ commonly requires $\mathcal{O}(n^3)$ operations. Luckily, (Hammond et al., 2011) provides us a method to fast approximate the spectral graph wavelet via Chebyshev polynomials. Let $s$ be the fixed scaling parameter in the heat filter kernel $g_s(\lambda) = e^{-\lambda s}$ and $M$ be the degree of the Chebyshev polynomial approximations for the scaled wavelet (Larger value of $M$ yields more accurate approximations but higher computational cost in opposite), the graph wavelet is given by

$$\psi_s = \frac{1}{2}c_{0,s} + \sum_{i=1}^{M} c_{i,s}T_i(\tilde{L}),$$

$$c_{i,s} = \frac{2}{\pi}\int_0^\pi \cos i\theta e^{-s(\cos\theta+1)}d\theta = 2e^{-s}J_i(-s) \qquad (12)$$

where $\tilde{L} = \frac{2}{\lambda_{\max}}L - I_n$ and $J_i(-s)$ is the Bessel function of the first kind. The proof can be referred to (Hammond et al., 2011). With this Chebyshev polynomial approximation, the computational cost of spectral graph wavelets is decreased to $\mathcal{O}(M\|E\| + M \times n)$. Due the real world graphs are usually sparse, this computational difference can be very significant.

## 4 Experiments

### 4.1 Datasets

We evaluate our proposed *OctGCN* on semi-supervised node classification task. The experimental setup is closely followed (Yang et al., 2016; Kipf & Welling, 2017). Statistical overview of

Table 2: Experimental results (in percent) on semi-supervised node classification.

| Model | Citeseer | Cora | Pubmed |
|---|---|---|---|
| LP (Zhu et al., 2003) | 45.3 | 68.0 | 63.0 |
| ICA (Lu & Getoor, 2003) | 69.1 | 75.1 | 73.9 |
| ManiReg (Belkin et al., 2006) | 60.1 | 59.5 | 70.7 |
| SemiEmb (Weston et al., 2012) | 59.6 | 59.0 | 71.1 |
| DeepWalk (Perozzi et al., 2014) | 43.2 | 67.2 | 65.3 |
| Planetoid (Yang et al., 2016) | 64.7 | 75.7 | 77.2 |
| Spectral CNN (Bruna et al., 2014) | 58.9 | 73.3 | 73.9 |
| ChebyNet (Defferrard et al., 2016) | 69.8 | 81.2 | 74.4 |
| Vanilla GCN (Kipf & Welling, 2017) | 70.3 | 81.5 | 79.0 |
| GWNN (Xu et al., 2019a) | 71.7 | 82.8 | 79.1 |
| LNet (Liao et al., 2019) | $66.2 \pm 1.9$ | $79.5 \pm 1.8$ | $78.3 \pm 0.3$ |
| AdaLNet (Liao et al., 2019) | $68.7 \pm 1.0$ | $80.4 \pm 1.1$ | $78.1 \pm 0.4$ |
| SGC (Wu et al., 2019) | $71.9 \pm 0.1$ | $81.0 \pm 0.0$ | $78.9 \pm 0.0$ |
| MoNet (Monti et al., 2017) | — | $81.7 \pm 0.5$ | $78.8 \pm 0.3$ |
| GAT (Veličković et al., 2018) | $\mathbf{72.5} \pm 0.7$ | $83.0 \pm 0.7$ | $79.0 \pm 0.3$ |
| GIN (Xu et al., 2019b) | $66.1 \pm 0.9$ | $77.6 \pm 1.1$ | $77.0 \pm 1.2$ |
| DGI (Velickovic et al., 2019) | $71.8 \pm 0.7$ | $82.3 \pm 0.6$ | $76.8 \pm 0.6$ |
| *OctGCN* (this paper) | $72.1 \pm 0.2$ | $\mathbf{83.5} \pm 0.2$ | $\mathbf{80.5} \pm 0.3$ |

datasets is given in Table 1. Three real-world datasets are chosen as benchmarks: Citeseer, Cora and Pubmed (Sen et al., 2008). In these citation networks, nodes are documents with corresponding bag-of-words features and edges are citation links. Label rate denotes the ratio of labeled nodes fetched in training process. We keep the label rate consistent with the classic public split, which is 20 labeled nodes per class in each dataset for training. Meantime, the test set contains 1000 labeled samples for prediction accuracy evaluation, and the validation set includes 500 labeled samples for determining hyper-parameters.

## 4.2 BASELINES

We first compare against traditional baselines, *i.e.*, label propagation (LP) (Zhu et al., 2003), iterative classification algorithm (ICA) (Lu & Getoor, 2003), manifold regularization (ManiReg) (Belkin et al., 2006), semi-supervised embedding (SemiEmb) (Weston et al., 2012), skip-gram based graph embeddings (DeepWalk) (Perozzi et al., 2014) and Planetoid (Yang et al., 2016).

Then we compare the most recent and state-of-the-art baselines from both spectral and spatial graph neural networks, since they are shown effective for semi-supervised settings. For spectral approaches based on convolution theorem, we compare our *OctGCN* with the Spectral CNN (Bruna et al., 2014), ChebyNet (Defferrard et al., 2016), Vanilla GCN (Kipf & Welling, 2017), GWNN (Xu et al., 2019a), LNet/AdaLNet (Liao et al., 2019) and SGC (Wu et al., 2019). For spatial based methods, we select the MoNet (Monti et al., 2017), GAT (Veličković et al., 2018), GIN (Xu et al., 2019b) and DGI (Velickovic et al., 2019) as comparisons.

## 4.3 EXPERIMENTAL SETUP

For all experiments, a 2-layer network of our model is constructed using TensorFlow (Abadi et al., 2015) with 64 hidden units. We train our model utilizing the Adam optimizer (Kingma & Ba, 2014) with an initial learning rate $lr = 0.01$. We terminate training if validation accuracy does not improve for 100 consecutive steps, and most runs finish in less than 200 steps as expected. We initialize the weights matrix following (Glorot & Bengio, 2010), employ $5 \times 10^{-4}$ L2 regularization on weights and dropout input and hidden layers to prevent overfitting (Srivastava et al., 2014).

For hyper-parameters for constructing wavelets $\psi_s$, we adopt the selection of the scaling parameter $s$ and sparseness threshold $t$ (the elements of $\psi_s$ are set to 0 when smaller than $t$) as in (Xu et al., 2019a), *i.e.*, $s = 0.7$ $t = 1 \times 10^{-5}$ for Citeseer, $s = 1.0$ $t = 1 \times 10^{-4}$ for Cora and $s = 0.5$ $t = 1 \times 10^{-7}$ for Pubmed, since both smaller $s$ and $t$ are shown not sensitive to datasets. For

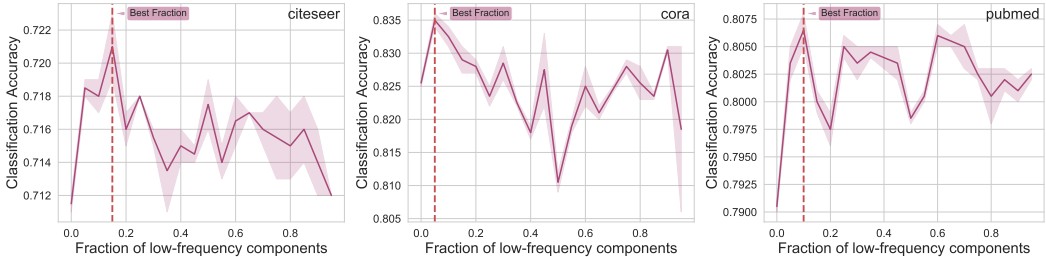

Figure 2: The performance of learned *OctGCN* w.r.t the proportion of low-frequency components. The best fraction is marked with the red vertical line.

Table 3: Learned weights $\alpha_L$ and $\alpha_H$ of *OctGCN* for low and high frequency w.r.t the best fraction of low frequency components $d/n$ (number followed after the name of datasets).

| Dataset | Citeseer (15%) | | Cora (5%) | | Pubmed (10%) | |
|---|---|---|---|---|---|---|
| Octave filter weights | $\alpha_L$ | $\alpha_H$ | $\alpha_L$ | $\alpha_H$ | $\alpha_L$ | $\alpha_H$ |
| Learned value | **0.838** | 0.162 | **0.722** | 0.278 | **0.860** | 0.140 |

Table 4: The mean Silhouette Coefficient of learned samples. Larger is better.

| Dataset | Citeseer | | | Cora | | | Pubmed | | |
|---|---|---|---|---|---|---|---|---|---|
| Model | Vanilla GCN | GWNN | *OctGCN* | Vanilla GCN | GWNN | *OctGCN* | Vanilla GCN | GWNN | *OctGCN* |
| Silhouette score | 0.038 | 0.050 | **0.083** | 0.119 | 0.153 | **0.220** | 0.110 | 0.130 | **0.171** |

the only hyper-parameter of *OctGCN*, the optimal proportion $d/n$ of low-frequency components for each dataset, is determined through grid search and studied in next Section. The weights of low- and high-frequency components $\alpha_L$ and $\alpha_H$ are both initialized with 1 and learned automatically. In experiments, Max-pooling is chosen to demonstrate the importance the low-frequency components.

### 4.4 EXPERIMENTAL RESULTS

#### 4.4.1 PERFORMANCE OF *OctGCN* ON NODE CLASSIFICATION

In Table 2, we demonstrate how our model performs on public splits taken from (Yang et al., 2016). The results of baselines are strictly consistent with the numbers from literature. With the limited information given in semi-supervised learning, We achieve a average test accuracy of 72.1%, 83.5%, and 80.5% on Citeseer, Cora and Pubmed, respectively. As *OctGCN* learned the octave feature representations for graph in spectral domain, it can demonstrate the meaningful information extracted from the underlying "true signal" from low-frequency over high-frequency. This is the main reason that explains why *OctGCN* outperforms other baseline methods.

#### 4.4.2 ANALYSIS ON INTERPRETABILITY

In Figure 2, how the proportion $d/n$ of low-frequency components affect the performance is studied. We fine-tune the proportion in a range of $\{0\%, 5\%, \cdots, 95\%\}$. The best proportion of low-frequency components are 15%, 5%, and 10% for Citeseer, Cora and Pubmed, respectively. The learned weights of low- and high-frequency components $\alpha_L$ and $\alpha_H$ w.r.t the best proportion for each dataset are demonstrated in Table 3, accordingly. It's clearly to note that the small proportion of low-frequency components are essential to the learning octave feature representation. The results are in line with the importance of low-frequency in GSP and bring interpretability to the nature of GCNs.

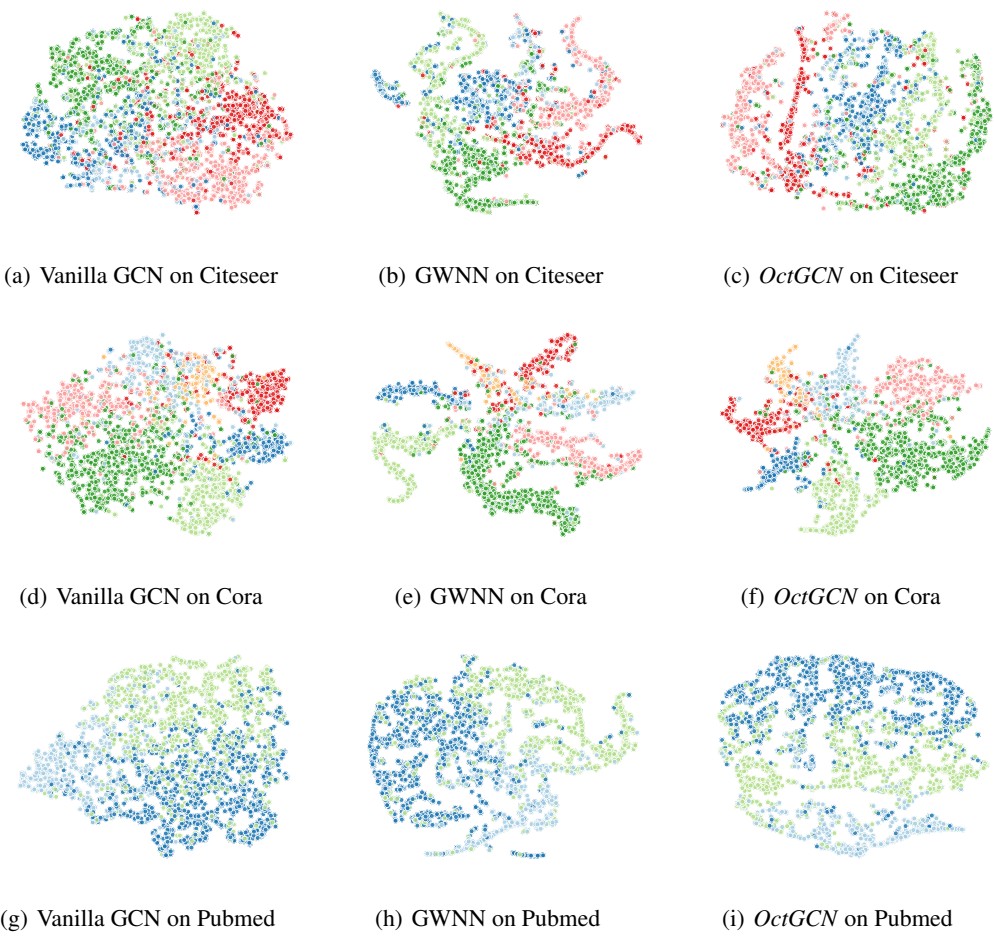

(a) Vanilla GCN on Citeseer   (b) GWNN on Citeseer   (c) *OctGCN* on Citeseer

(d) Vanilla GCN on Cora   (e) GWNN on Cora   (f) *OctGCN* on Cora

(g) Vanilla GCN on Pubmed   (h) GWNN on Pubmed   (i) *OctGCN* on Pubmed

Figure 3: The t-SNE visualization of *OctGCN* comparing with spectral convolution based baselines. Each color corresponds to a different class that the embeddings belongs to.

### 4.4.3 T-SNE VISUALIZATION OF LEARNED EMBEDDINGS

Table 4 presents the mean Silhouette Coefficient (Rousseeuw, 1987) over all learned samples, larger the silhouette score is, better the clustering performs. We choose two representative baseline methods, *i.e.*, Vanilla GCN (Kipf & Welling, 2017) and GWNN (Xu et al., 2019a) for comparison. We can indicate that *OctGCN* achieves the best quality of embeddings. Figure 3 depicts the t-SNE visualization (Maaten & Hinton, 2008) of learned embeddings on all three citation datasets. We can visualize the local and sparse property of spectral graph wavelets that utilized in GWNN and *OctGCN*. Further, the intersections of different classes are more separated in the results our *Oct-GCN*, since the octave feature embeddings learned from our model tend to capture the importance information in low-frequency components and effectively alleviate the noise from high-frequency.

## 5 CONCLUSION

In this paper, we propose *OctGCN*, a novel spectral-based graph convolutional neural network to learn the representation of graph with respect to different frequency components. By distinct design of filters for low- and high-frequency, our model can effectively capture the octave feature representations and enhance the interpretability of GCNs. To the best of our knowledge, this is the first attempt on octave convolution for graphs. An interesting direction for future work is to extend the definition of octave convolution from spectral domain to spatial domain, in order to pursue more efficient architectures for learning with graphs.

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

Table 5: The average gap of MAD and INR between the low- and high- frequency components in benchmark datasets. Smaller MAD indicates better smoothness and larger INR indicates that the frequency components contain richer information.

| Dataset | Citeseer | Cora | Pubmed |
|---|---|---|---|
| $MAD_L - MAD_H$ | $-3.7 \times 10^{-4}$ | $-9.0 \times 10^{-5}$ | $-1.0 \times 10^{-5}$ |
| $INR_L - INR_H$ | 0.018 | 0.002 | 0.005 |

Felix Wu, Amauri H. Souza, Tianyi Zhang, Christopher Fifty, Tao Yu, and Kilian Q. Weinberger. Simplifying graph convolutional networks. In *ICML 2019 : Thirty-sixth International Conference on Machine Learning*, pp. 6861–6871, 2019.

Zonghan Wu, Shirui Pan, Fengwen Chen, Guodong Long, Chengqi Zhang, and Philip S Yu. A comprehensive survey on graph neural networks. *arXiv preprint arXiv:1901.00596*, 2019.

Bingbing Xu, Huawei Shen, Qi Cao, Yunqi Qiu, and Xueqi Cheng. Graph wavelet neural network. *international conference on learning representations*, 2019a.

Keyulu Xu, Weihua Hu, Jure Leskovec, and Stefanie Jegelka. How powerful are graph neural networks. In *ICLR 2019 : 7th International Conference on Learning Representations*, 2019b.

Zhilin Yang, William W. Cohen, and Ruslan Salakhutdinov. Revisiting semi-supervised learning with graph embeddings. In *ICML 2016*, pp. 40–48, 2016.

Liang Yao, Chengsheng Mao, and Yuchen Luo. Graph convolutional networks for text classification. *AAAI 2019 : Thirty-Third AAAI Conference on Artificial Intelligence*, 33:7370–7377, 2019. URL https://academic.microsoft.com/paper/2962946486.

Ziwei Zhang, Peng Cui, and Wenwu Zhu. Deep learning on graphs: A survey. *arXiv preprint arXiv:1812.04202*, 2018.

Xiaojin Zhu, Zoubin Ghahramani, and John D. Lafferty. Semi-supervised learning using gaussian fields and harmonic functions. In *ICML 2003*, pp. 912–919, 2003.

# 6 APPENDIX

## 6.1 LOW-FREQUENCY COMPONENTS IMPLY BETTER SMOOTHNESS AND INFORMATION-TO-NOISE RATIO

The different importance of low- and high-frequency components of graphs that contributes to the learning of modern GNNs is observed recently (Donnat et al., 2018; Maehara, 2019). Concretely, the low-frequency components in graphs usually indicate smooth varying signals which can reflect the locality property (the neighbor nodes trend to be similar with each other) in graphs, thus they capture more information than the high-frequency components and should be more beneficial for the representational learning as revealed in our experiments. To this end, we investigate two measures in (Chen et al., 2019): Mean Average Distance (MAD) and Information-to-Noise Ratio (INR) w.r.t different spectrum basis. Different spectrum basis has a corresponding node in graph, which we refer as low- and high- frequency node.

MAD reflects the smoothness of node representation. Given a node $v$ and the feature $\mathbf{x}_v$ on it as signal, we take its eigenvector $\mathbf{u}_v$ to multiply $\mathbf{x}_v$ as the transformed feature $\tilde{\mathbf{x}}_v$ in spectral domain. Then the MAD of $v$ is calculated by taking the average of cosine distances between the transformed feature of $v$ and that of its 1-hop neighbors. The lower MAD indicates better smoothness. INR is defined as the proportion of nodes from the same class as $v$ through the 1-hop neighborhood of $v$. INR reflects the information contained in nodes and the higher INR indicates the richer information. For each dataset, we compute the average MAD and INR of top-50% low-frequency nodes ($MAD_L$ and $INR_L$) and rest high-frequency nodes ($MAD_H$ and $INR_H$) and report the gap between low- and high- frequency nodes as in Table 5.

We can observe that the low-frequency components in real graphs indeed are smoother than that of high-frequency (lower MAD). Meanwhile, they have higher INR, which indicates they contain richer information. In the vein, we argue that the low-frequency components may carry more information than that of the high-frequency components and should be more beneficial for the representational learning. Therefore, comparing with all other spectral-based methods that treat both low- and high-frequency components identically during training, our proposed octave convolutional structure model could gain more from this octave nature existing in graphs.

