# OpenReview forum: "Octave Graph Convolutional Network"
_ICLR.cc/2020/Conference — Reject_

### Official Review · AnonReviewer2 · 2019-10-25
**Official Blind Review #2**

**Rating:** 6

**Review:**

Despite reading the paper multiple times,  I am not sure I have the background to know whether what is written is significant or not. I'm aware of work on general semi-supervised learning and see the much better performance of this approach compared to things like label propagation, but cannot say for sure whether the idea is novel/significant.

One q for authors -- I don't understand the core component of the proposal, is the key ingredient that have different weighting between low vs high that causes the better performance on tasks ? or is that we have less dependencies across variables (as reflected in the computational costs) that gets the better performance ?

**Experience Assessment:**

I do not know much about this area.

**Review Assessment: Checking Correctness Of Derivations And Theory:**

I did not assess the derivations or theory.

**Review Assessment: Checking Correctness Of Experiments:**

I did not assess the experiments.

**Review Assessment: Thoroughness In Paper Reading:**

I read the paper at least twice and used my best judgement in assessing the paper.

---

> ### Author Response · Authors · 2019-11-14
> **Response to Official Blind Review #2**
>
> Thank you for your earnest manner on review and we appreciate it very much!
> Indeed, both the different weighting between low vs high-frequency components and fewer dependencies across variables contribute to better performance.
> 1. By assigning different weights between low vs high, the different importance of low vs high-frequency components existing in graphs is captured for the first time in our proposed model. For more details of our motivation, please kindly refer to the response to Reviewer #3.
> 2. Along with the octave convolutional operations, we further reduce the parameters in our model as described in Section 3.4. Since the amount of training data is extremely limited in semi-supervised learning, the reduction of parameters alleviates the dependencies across variables and makes the representational learning have better performance.

---

### Official Review · AnonReviewer3 · 2019-10-27
**Official Blind Review #3**

**Rating:** 3

**Review:**

This paper proposes to use octave convolution to learn a representation of a graph.
Typically, a learning on a graph is done either in a spatial domain or in a spectral domain.
A spectral domain based approach uses a eigenvalue decomposition form of a graph Laplacian (a symmetric matrix) and learning a filter that acts on the eigenvalue of a graph Laplacian while preserving eigenvectors of a graph Laplacian.
This architecture is called the graph convolutional network on a spectral domain.

This paper's main contribution is to adapt octave convolutional network's architecture to the usual graph convolutional network. While I believe that this is the first work on applying the idea behind octave convolutional network architecture, separating low and high frequency component in the learning stage, to graph convolutional network architecture, I cannot see a good motivation on why this architecture is good for learning on a graph.
A comprehensive study in the paper shows a better performance gain compared to the existing method, but it would be better if the gains were substantial or the authors presented a good motivation on why this architecture is good in some cases.

Overall, I think the paper is well-written, but I would suggest to present more meaningful justification why and when the octave GCN is better than the GCN.

**Experience Assessment:**

I have read many papers in this area.

**Review Assessment: Checking Correctness Of Derivations And Theory:**

I assessed the sensibility of the derivations and theory.

**Review Assessment: Checking Correctness Of Experiments:**

I assessed the sensibility of the experiments.

**Review Assessment: Thoroughness In Paper Reading:**

I made a quick assessment of this paper.

---

> ### Author Response · Authors · 2019-11-14
> **Response to Official Blind Review #3**
>
> Thank you for your recognition of the novelty of our works and valuable comments. We are sorry that our motivation is not specified sufficiently. Here we elaborate our motivation by example in the following.
> The octave nature in computer vision usually denotes that the low-frequency components describe the smoothly changing structure (e.g. background) and high-frequency components describe the rapidly changing details (e.g. outlines) [A1].
>
> Similarly, the low- and high- frequency components in the nature of graphs also exhibit the different importance that contributes to the learning of modern GNNs [A2]. Concretely, we observe that the low-frequency components in graphs usually indicate smooth varying signals which can reflect the locality property (the neighbor nodes trend to be similar to each other) in graphs. To validate this observation, we investigate two measures in [A3]: Mean Average Distance (MAD) and Information-to-Noise Ratio (INR) w.r.t different spectrum basis. Different spectrum basis has a corresponding node in graph, which we refer as low- and high- frequency node.
>
> MAD reflects the smoothness of the transformed graph signal on node. Given a node $v$ and the feature $\mathbf{x}_{v}$ on it as signal, we take its eigenvector $\mathbf{u}_{v}$ to multiply $ \mathbf{x}_{v}$ as the transformed feature $ \tilde{\mathbf{x}_{v}}$ in spectral domain. Then the MAD of $v$ is calculated by taking the average of cosine distances between the transformed feature of $v$ and that of its 1-hop neighbors. The lower MAD indicates better smoothness.
>
> INR is defined as the proportion of nodes from the same class as $v$ through the 1-hop neighborhood of $v$. The higher INR indicates richer information.
>
> For each dataset, we compute the average MAD and INR of top-50% low-frequency nodes ($MAD_L$/$INR_L$) and high-frequency nodes ($MAD_H$/ $INR_H$) and report the gap between low- and high- frequency nodes.
> Dataset | Citeseer | Cora | Pubmed
> $MAD_L$ – $MAD_H$ | $-3.7 \times 10^{-4}$ | $ -9.0\times 10^{-5}$ | $ -1.0\times 10^{-5}$
> $INR_L$ – $INR_H$ | $0.018$ | $0.002$ | $0.005$
> We can observe that the low-frequency components in real graphs are smoother than those of high-frequency (lower MAD). Meanwhile, they have higher INR, which indicates they contain richer information. In the vein, we argue that the low-frequency components may carry more information than that of the high-frequency components and should be more beneficial for representational learning. Therefore, comparing with current GNNs, that treat the low- and high- frequency components identically during training, our octave convolutional structure model could gain more from this octave nature existing in graphs. We updated this demonstration in the submission.
>
> [A1] Chen, Yunpeng, et al. “Drop an Octave: Reducing Spatial Redundancy in Convolutional Neural Networks with Octave Convolution.” ArXiv Preprint ArXiv:1904.05049, 2019.
> [A2] Nt, Hoang, and Takanori Maehara. “Revisiting Graph Neural Networks: All We Have Is Low-Pass Filters.” ArXiv Preprint ArXiv:1905.09550, 2019.
> [A3] Chen, Deli, et al. "Measuring and Relieving the Over-smoothing Problem for Graph Neural Networks from the Topological View." arXiv preprint arXiv:1909.03211 (2019).

---

### Official Review · AnonReviewer1 · 2019-10-31
**Official Blind Review #1**

**Rating:** 3

**Review:**

This paper extends the previous graph wavelet neural network with separate computation for low frequency and high frequency part. The idea is to bring the octave convolution in vision to the graph domain. Experiments on three benchmark node classification tasks show comparable performances as previous methods.

Overall the paper is written in a coherent and self-contained way, where the paper clearly states the related work and the contribution of this newly proposed work. Also it is interesting to see that normalizing the diagonal filter, tying the weights for low/high frequency parts would make the generalization better. However, there are several major concerns with the paper:

1. The contribution is somewhat limited. The main component is based on GWNN. While the GWNN itself takes the full spectrum of basis already, the formulation in (8) and (9) should somehow capture the same information. I think the paper spends too much content on the reviews, while lacks the intuition or theoretical explanations of the proposed formulation.

2. Having marginal improvement on the three benchmarks is not that interesting. Also given the results are mixed with other baselines, I think more experiments (e.g., on large graphs, or graph-level supervised tasks, etc.) are necessary to demonstrate the empirical gain using the proposed formulation.

3. Regarding the experiments, is it true that d/n=0 and d/n=1 should have exactly the same results?

4. In Figure 2, I guess d/n=0 should be reduced to the GWNN. But it seems the performance is different than GWNN on citeseer and cora. Why there’s such inconsistency, or did I miss anything?

**Experience Assessment:**

I have published in this field for several years.

**Review Assessment: Checking Correctness Of Derivations And Theory:**

I assessed the sensibility of the derivations and theory.

**Review Assessment: Checking Correctness Of Experiments:**

I assessed the sensibility of the experiments.

**Review Assessment: Thoroughness In Paper Reading:**

I read the paper at least twice and used my best judgement in assessing the paper.

---

> ### Author Response · Authors · 2019-11-14
> **Response to Official Blind Review #1**
>
> Thank you for the kind review and feedback and we sincerely appreciate it. In the following, we address the concerns point by point.
> Q1: The contribution of our paper.
>
> A1: We agree that the main structure of the proposed model is derived from GWNN and Spectral CNN [A1]. However, this paper is not intended to propose an extension of GWNN. Our intuition is behind the octave nature of graphs, that is, the low- and high- frequency components may contribute differently to the learning of GNNs [A2].  To better capture the different importance of low- and high- frequency components, we propose to learn two different filters $F_{L}$ and $F_{H}$ for graph convolution (9), respectively. In this paper, we choose GWNN as the backbone to demonstrate the effectiveness of our method due to the local and sparse property of spectral wavelet basis, which is beneficial to the learning. It’s worth to point out that the low- and high- frequency filters can be easily adapted to another spectral basis, such as Fourier basis, and extended to other GNN backbones. We included more explanation of the intuition of the proposed formulation in the revised version.
>
> Q2: More experiments.
>
> A2: The benchmark datasets that we used in the experiments are well-studied and well-tuned in the graph learning field. The experiments are standardized and fair for comparisons with the mixed baselines from both spatial-based and spectral-based GNN families. The improvement on the performance could legitimately demonstrate the empirical gain with our proposed model.
> We agree that more experiments, such as on larger graphs or various tasks, could be more interesting. However, the more important point of our experiments is that the proposed model can capture the importance of low-frequency components (Figure 2 & Table 3) and learn the better representation of the graph in our experiments (Figure 3 & Table 4). The results also validate our intuition of octave convolutional structure.
>
> Q3: Regarding the experiments, is it true that d/n=0 and d/n=1 should have exactly the same results?
>
> A3: Yes, it is true that d/n=0 and d/n=1 have exactly the same results, since full spectrum of basis is considered with same filter in both situations.
>
> Q4: In Figure 2, I guess d/n=0 should be reduced to the GWNN. But it seems the performance is different than GWNN on citeseer and cora. Why there’s such inconsistency, or did I miss anything?
>
> A4: When d/n=0, our model is still different from the GWNN. The diagonal filter in GWNN is learned by N different parameters, while we further reduce the parameters in the filters for low- and high- frequency components $F_{L}$ and $F_{H}$ with the identical parameters on the diagonal controlled by two weight parameters $\alpha_{L}$ and $\alpha_{H}$. Therefore, the diagonal filter of our model is learned by N identical weight parameters when d/n=0. Since the graph-based semi-supervised learning might prohibit the parameter learning due to the limited amount of training data, this reduction in parameters of our model could be essential for the representational learning and explain the better performance of our model.
>
> [A1] Bruna, Joan, et al. “Spectral Networks and Locally Connected Networks on Graphs.” ICLR 2014 : International Conference on Learning Representations (ICLR) 2014, 2014.
> [A2] Nt, Hoang, and Takanori Maehara. “Revisiting Graph Neural Networks: All We Have Is Low-Pass Filters.” ArXiv Preprint ArXiv:1905.09550, 2019.

---

### Decision · Program_Chairs · 2019-12-19

**Decision:**

Reject

**Comment:**

Two reviewers are negative on this paper while the other one is slightly positive. Overall, the paper does not make the bar of ICLR and thus a reject is recommended.